# Early Application of Ipsilateral Cathodal-tDCS in a Mouse Model of Brain Ischemia Results in Functional Improvement and Perilesional Microglia Modulation

**DOI:** 10.3390/biom12040588

**Published:** 2022-04-17

**Authors:** Laura Cherchi, Daniela Anni, Mario Buffelli, Marco Cambiaghi

**Affiliations:** Department of Neurosciences, Biomedicine and Movement Sciences, University of Verona, 37134 Verona, Italy; laura.cherchi@univr.it (L.C.); daniela.anni@univr.it (D.A.)

**Keywords:** non-invasive brain stimulation, focal ischemia, microglia, motor recovery

## Abstract

Early stroke therapeutic approaches rely on limited options, further characterized by a narrow therapeutic time window. In this context, the application of transcranial direct current stimulation (tDCS) in the acute phases after brain ischemia is emerging as a promising non-invasive tool. Despite the wide clinical application of tDCS, the cellular mechanisms underlying its positive effects are still poorly understood. Here, we explored the effects of cathodal tDCS (C-tDCS) 6 h after focal forelimb M1 ischemia in Cx3CR1^GFP/+^ mice. C-tDCS improved motor functionality of the affected forelimb, as assessed by the cylinder and foot-fault tests at 48 h, though not changing the ischemic volume. In parallel, histological analysis showed that motor recovery is associated with decreased microglial cell density in the area surrounding the ischemic core, while astrocytes were not affected. Deeper analysis of microglia morphology within the perilesional area revealed a shift toward a more ramified healthier state, with increased processes’ complexity and a less phagocytic anti-inflammatory activity. Taken together, our findings suggest a positive role for early C-tDCS after ischemia, which is able to modulate microglia phenotype and morphology in parallel to motor recovery.

## 1. Introduction

Stroke is one of the primary causes of disability, with a high rate of mortality worldwide. Pharmacological thrombolysis and mechanical thrombectomy are currently the only approved treatments for this pathology; nevertheless, their strict eligibility criteria and short time window, ranging up to 6 h after the onset of stroke symptoms, limit their therapeutic use [1,2,3]. Therefore, there is a pressing need to develop alternative effective therapies for enhancing functional stroke recovery and reducing patients’ disability, one of the major challenges for both preclinical and clinical stroke research at present.

It has been widely demonstrated that the very early phases of an ischemic event (minutes up to few hours) are the best time windows for intervention. Immediately after focal cerebral ischemia, excitotoxicity overactivates several downstream signaling pathways, leading to neuronal and glial cell injury, which in turn triggers a sequence of pathogenic events that contribute to increased tissue loss and motor functional neurological deficits [4,5,6]. The inflammatory response, despite its well-known positive effect in structural and functional brain recovery after stroke, strongly contributes to a secondary acute brain damage in the early phases after ischemic injury [7]. Within minutes after brain ischemia, the brain’s resident immunological macrophage-like cells, known as microglia cells, are activated toward a phagocytotic phenotype and proliferate in the regions surrounding the ischemic core. Once activated, these cells change their morphology from ramified to ameboid with retracted processes and an enlarged cell body [8,9,10], releasing a series of inflammatory mediators that cause a worsening of tissue damage [5,10]. Thus, the post-ischemic inflammatory processes strongly depend on microglial response which, in addition to the pro-inflammatory polarization, exhibit a protective anti-inflammatory response, characterized by the expression of different cell surface proteins and production of different anti-inflammatory molecules [5,7,10,11]. Together with microglia, astrocytes have been shown to quickly activate and strongly influence the prognosis of ischemic brain outcomes [11].

In this contest, the non-invasive neuromodulatory technique transcranial direct current stimulation (tDCS) is emerging as a valid approach to counteract the inter-hemispheric functional imbalance in the chronic, but also in the early phases, after stroke [12], with strong evidence in both animal [13,14] and human studies [15,16,17]. Besides the polarity-dependent direct effect on neuronal excitability, tDCS might be an add-on tool capable of modulating the ischemic pathophysiology, including the inflammatory response characterizing the acute/subacute phases [5,18], due to its impact on both astrocytes [19] and microglia cells [14,20,21]. Moreover, in our previous work, we showed that application of cathodal but not anodal tDCS during the very early phases of ischemic stroke has neuroprotective and better functional outcomes in a middle cerebral artery occlusion (MCAO) mouse model [14].

Thus, our aim was to study the effects of the acute application (6 h after ischemia) of cathodal tDCS (C-tDCS) on motor functionality and microglia activation in a mouse model of ischemic stroke.

For this purpose, we induced a focal ischemic injury via photothrombosis in the M1 region of Cx3CR1^GFP/+^ mice and, six hours later, we applied an intermittent C-tDCS protocol. Behavioral, immunohistochemical and microglial morphological analysis showed that C-tDCS increased forelimb motor function and reduced microglia density in the perilesional area, evoking a more resting microglial and anti-inflammatory phenotype.

## 2. Materials and Methods

### 2.1. Animals

A total of 36 heterozygous Cx3CR1^GFP/+^ mice (10–12 weeks, 25–30 g) were used for this study. Animals were housed in conventional cages in groups of two to four per cage and were kept in a humidity and temperature-controlled room with a 12-h light–dark cycle and ad libitum access to food and water. All procedures were performed in accordance with the EU Council Directive 2010/63/EU on the protection of animals used for scientific purposes and were approved by the Italian Ministry of Health.

### 2.2. Experimental Design

Two days before photothrombotic lesion induction, animals performed the cylinder test (baseline) and were trained to the rotarod. On the next day, mice were subjected to gridwalk and rotarod tests (baseline). According to the results obtained from the cylinder test, we determined the forelimb preference for each mouse and we performed the photothrombotic stroke induction or no-stroke (NS) procedure on the contralateral hemisphere. Immediately after, the electrode for tDCS was implanted. Two mice died after photothrombotic induction (one immediately after and one after 2 h); a post mortem analysis revealed no obvious cause of death such as hemorrhage, thus death probably resulted from ischemia (mortality rate of 5.5%). Six hours after photothrombosis, cathodal tDCS (C-tDCS) or no stimulation (sham) were applied. The animals (*n* = 34) were allowed to recover for two days, after which they were subjected to a post-ischemia cohort of motor tests in the following order: cylinder test, grid walking test, open field test and rotarod test (intervals of at least 2 h). Later, animals were intracardially perfused and their brain was extracted for histological analysis (Figure 1A–C).

After baseline tests, animals were randomly divided in four experimental groups: NS/sham = 8, NS/C-tDCS = 8, NS/sham = 9, stroke/C-tDCS = 9. For each condition, a subgroup (*n* = 4) was used for histological analysis.

### 2.3. Photothrombotic Stroke Induction and Epicranial Electrode Implantation

Mice were anaesthetized with ketamine-xylazine (80 mg/kg and 5 mg/kg, respectively, i.p.) and placed on the stereotaxic frame. The cranium was exposed and M1 forelimb area (+0.4 mm AP and ± 1.6 mm ML from Bregma) illuminated for 12 min by a green light laser (33 mW), filtered through a 10× objective for five minutes following Rose Bengal injection (200 μL of 10 mg/mL, i.p.). In sham animals, the same procedure without laser illumination was performed.

After photothrombotic stroke induction, a tubular plastic jack (inner area = 4.5 mm^2^) was cemented epicranially and centered on the previously illuminated area [22]. Mice were then allowed to recover from anesthesia and returned to their home cages.

### 2.4. tDCS Stimulation

Six hours after the photothrombotic lesion, cathodal current was applied with a protocol consisting of 20′ ON 20′ OFF 20′ ON 250 μA tDCS in awake animals [14]. The tubular electrode support was filled with saline solution, while the counter electrode applied to the shaven ventral thorax was soaked with saline solution and hydrosoluble gel. Sham animals underwent the same procedure with the tDCS stimulator switched off.

### 2.5. Behavioural Analysis

A cohort of behavioral tests was performed. Before performing all tests, mice were acclimatized to the testing room for 1 h.

#### 2.5.1. Cylinder Test

The cylinder test was used to evaluate the spontaneous asymmetric forelimb use during events of vertical exploration in a confined environment. Mice were individually placed into a glass cylinder (9 cm diameter, 20 cm height) and allowed to explore for 5 min, while a camera was placed below the cylinder. At the end of the test, the animal was returned to its home cage and the cylinder was cleaned with 70% ethanol. Preferences in forelimb use were assessed by counting the number of times the mice left each forepaw at the bottom of the cylinder to push its body weight to a vertical position, touched the cylinder wall with a single forelimb during exploratory activity pushing itself from a vertical position to a sitting position and left a single forelimb on the bottom of the cylinder glass to sustain its bodyweight down to a sitting position. Movements performed with both forelimbs at the same time were not considered. A baseline evaluation was executed to reduce the bias of individual variation and to determine the pre-operatory preferential use of the forelimbs, allowing designation of the preferred forelimb as ipsilateral to the hemisphere subjected to photothrombotic stroke.

The asymmetry index was calculated by a blinded experimenter according to the formula:Asymmetry Index = (C_ipsi_/C_ipsi_ + C_contra_) × 100 − (C_contra_/C_ipsi_ + C_contra_) × 100

#### 2.5.2. Gridwalk Test

The gridwalk test was used to evaluate the coordination of mice and a possible unilateral motor impairment. The apparatus consisted of an elevated and levelled grid (25 × 25 cm) composed of 1 × 1 cm metal wire openings (0.8 mm diameter) with a camera placed below the grid. Briefly, mice were individually placed on the grid and set free to roam for 5 min while being video-recorded. At the end, the mouse was returned to its home cage and the apparatus was cleaned with 70% ethanol. The videos were analyzed offline, frame by frame, to assess the number of correct steps and foot faults. The experimenter was blinded to the experimental group. The latter were defined as steps not providing body support resulting in the falling of the forelimb into a grid hole. A baseline test was executed to normalize the post-operatory Foot Fault Index, which was calculated as follows [23]:Foot Faults% = 100 × #foot faults/#correct steps + #foot fault 

#### 2.5.3. Open Field Test

The open field test was used to assess spontaneous post-ischemic locomotor activity. This test was performed under dim light conditions (a red light was kept as the only source of illumination in the room). Mice were individually placed in the square arena (50 × 50 cm) surrounded by 40 cm high sidewalls and their activity recorded for 5 min using a camera, after which the mouse was returned to its home cage. Locomotion (distance traveled) was analyzed offline by a blinded experimenter.

#### 2.5.4. Rotarod Test

Motor coordination and balance was tested using the accelerating rotarod (4–40 rpm, in 5 min). The animals received a training session at constant velocity of 8 rpm until 60 s of constant running activity was reached. The test was executed at baseline and at post-operatory conditions. The experimenter was blinded to the experimental group. Mice were placed on a treadmill with a diameter of 3.5 cm, and the time until the animals either fell or performed two passive rotations was measured (latency in seconds). The maximum duration of a trial was set at 5 min. The test was performed twice with a 30 minutes’ inter-trial interval and means were used for statistical analysis. The rod was cleaned with 70% ethanol after each session.

#### 2.5.5. Immunohistochemistry

Following the behavioral session, mice were anaesthetized and transcardially perfused with cold phosphate-buffer saline (PBS) followed by cold 4% formalin solution. Brains were removed, post-fixed overnight and washed in PBS. Coronal sections of 35 μm thickness were obtained using a vibratome (Leica Systems, Wetzlar, Germany), conserved in PBS and protected from the light source.

For the assessment of brain injury, infarct volume was determined by staining one out of every three sections with Cresyl-violet. The ischemic areas were acquired using a 5× bright field microscope (Leica Systems, Wetzlar, Germany) and measured with ImageJ (https://imagej.nih.gov/ij/, accessed on 17 March 2022). For each animal, the lesion volume was determined by adding up all damage areas and multiplying the number by section thickness and by the spacing factor of 3. The lesion volume is given in mm^3^ and represents the mean ± standard error of all animals/group.

Moreover, three non-serial coronal sections (separated by at least 210 μm) were selected in four animal per condition and used for immunofluorescence staining for the astrocyte detection marker GFAP (1:300, Invitrogen anti-rat; cat.#130300), the M1 marker of active phagocytosis CD68 (1:200, Bio-Rad anti mouse, cat.# MCA1957), the M2 microglial marker CD206 (1:200, Abcam anti rabbit, cat.# ab64693) and the neural marker NeuN (1:800, Merck Millipore anti mouse, cat.# MAB377).

### 2.6. Image Acquisition, Analysis and 3D Microglia Reconstruction

Microglia, astrocyte and neural cells images were acquired using a fluorescence microscopy (Leica DM6000 B, Leica Mycrosistem, Wetzlar, Germany) with a HC PL FLUOTAR 20× air objective (0.50 numerical aperture (NA), Leica Mycrosystem, Wetzlar, Germany) and a digital camera. For immunofluorescence quantification, four regions of interest (ROIs; 332 × 445 μm) from each slice have been identified: (1) the infarcted area (core), (2) the boundary of the injury (perilesional), (3) a region 500 μm farther apart from the infarct area (healthy) and (4) a region in the contralateral motor cortex used to normalize the analysis or as a healthy region for microglial 3D analysis. Each ROI mean pixel intensity (optical density) was quantified with Image-J. Microglia GFP and GFAP and CD68 signals were measured as the percentage of fluorescent signal over the total area of the image using WEKA trainable segmentation plugin and then Analyse Particles function in ImageJ. Microglia and neural cells’ density was evaluated manually by counting the number of microglial GFP-positive and NeuN-positive cells whose soma was included in the ROI. Reported data are the ratio of each ROI over the contralateral. Of note, for CD206, the expression of which is mainly limited to the ischemic core, the percentage of CD206^+^ area was evaluated.

For 3D reconstruction of microglial cells, stacks of confocal images were acquired with a Leica SP5 confocal microscopy (Leica Microsystems, Wetzlar, Germany) at 1 μm intervals using a HC PL APO CS2 63× oil-immersion objective (1,4 numerical aperture (NA), Leica Microsystems, Wetzlar, Germany), 2.5× zoom and resolution of 1024 × 1024 pixels. Image stacks were imported into Imaris 9.1.2 software (Bitplane, Zurich, Switzerland) and subjected to morphological reconstruction. First, a Gaussian filter (smooth: 0.192 μm) was applied to remove image noise and then the microglial cell body and processes were reconstructed using, respectively, the “surface” and “filament tracer” function in Imaris. We selected individual cell bodies using the “Segment a region of interest” option in the surface algorithm, and we reconstructed the soma using a surface area detail of 0.192 μm with a manually adjusted threshold capable of covering the whole visible microglia volume. For the reconstruction of microglia processes, we selected the whole cell as a region of interest and, using Imaris Filament Tracer with no loops allowed, we set filament starting and seed points (the first defined by the soma diameter size and the second by the thinnest filament diameter in the slice mode). Next, based on filament diameter size, we adjusted the local threshold. False seed points and next false connections were manually removed. Data analyses were conducted in a blind manner in relation to experimental conditions.

### 2.7. Statistical Analysis

Statistical analysis among groups was performed using the Prism 8.3 software (GraphPad, La Jolla, CA, USA). All the data were normally distributed (Shapiro–Wilk normality test) and were compared with two-tailed unpaired t-test (comparisons between two groups) or with two-way ANOVA followed by a post hoc Tukey’s multiple comparisons test (comparisons among more than three groups). Differences among groups were considered statistically significant with a *p* < 0.05.

## 3. Results

### 3.1. Early Application of Cathodal tDCS after Stroke Improves Unilateral Motor Deficits

The photothrombotic stroke model resulted in a permanent occlusion of the blood flow in the light targeted area of the brain, centered on the forelimb M1. The volume of the observed lesion was not significantly different between non-stimulated (stroke/sham) and stimulated (stroke/C-tDCS) mice that received 40 min of C-tDCS 6 h after the ischemia onset (Figure 1D: stroke/sham 3.32 ± 0.74 mm^3^ vs stroke/C-tDCS 3.73 ± 0.92 mm^3^).

At 48 h after ischemia onset, the focal cortical lesion produced deficits in the forelimb preference (identified on the baseline cylinder test), as assessed on the cylinder and gridwalk tests (Figure 2A,B), with a significant main effect of stroke (F_1,30_ = 78.61, *p* < 0.0001 and F_1,30_ = 72.35, *p* < 0.0001, respectively). For both tests, the application of C-tDCS in the early phase after photothrombosis or in no stroke condition showed a significant stroke x stimulation interaction between baseline and post-stroke evaluation (F_1,30_ = 8.65, *p* = 0.0063 and F_1,30_ = 4.70, *p* = 0.0208, respectively). Post-hoc analysis with Tukey’s correction revealed that the C-tDCS resulted in a significant recovery of the affected forelimb function within the stroke group, in its voluntary use in the cylinder test (*p* = 0.028) and in the number of foot-faults in the gridwalk test (*p* = 0.046). No differences were observed between the no-stroke groups (*p* > 0.05).

Spontaneous locomotor activity and motor coordination estimating bilaterally symmetric dysfunction were measured with the open-field test and the rotarod. Both analyses found no differences among groups (Figure 2C,D), as also evident by no stroke x stimulation interaction in the total distance traveled in the OFT (F_1,30_ = 0.063, *p* < 0.79) and in the latency to fall from the rotating rod (F_1,30_ = 1.94, *p* < 0.17).

### 3.2. C-tDCS Has No Effect on Astrocytic Gliosis but Selectively Reduced Microglia Density in the Perilesional Area

Astrocytes are known to play a key role in the preservation of brain homeostasis, and they are promptly activated following brain damage. After a brain injury such as a stroke, astrocytes are characterized by a strong proliferation and concomitant expression of the structural protein GFAP. We analyzed the area of GFAP expression in the core, perilesional and healthy regions of the four different conditions (Figure 3), and we found a significant increase in the stroke main effect in all of them (F_1,12_ = 0.032, *p* = 0.91; F_1,12_ = 0.11, *p* = 0.79; F_1,12_ = 0.40, *p* = 0.69, respectively). However, no differences in the stroke x stimulation interaction were observed (F_1,12_ = 65.16, *p* = 0.0005; F_1,12_ = 79.87, *p* < 0.0001; F_1,12_ = 70.13, *p* = 0.0002, respectively for the core, perilesional and healthy ROIs).

To assess the effects of stroke and C-tDCS on microglia cells, we analyzed microglial density and the total GFP^+^ area in all three different regions, and we normalized values to the contralateral hemisphere (Figure 4). Cell counts of microglial cells in the core, perilesional and healthy ROIs revealed an effect of ischemia in all examined areas (F_1,12_ = 98.99, *p* < 0.0001; F_1,12_ = 31.66, *p* = 0.0001; F_1,12_ = 8.98, *p* = 0.011, respectively) but a significant stroke x stimulation interaction only in the perilesional region (F_1,12_ = 0.072, *p* = 0.79; F_1,12_ = 10.29, *p* = 0.0075; F_1,12_ = 0.042, *p* = 0.83, respectively for the core, perilesional and healthy ROIs). Post-hoc analysis with Tukey’s correction of the perilesional area indicated a significant decrease in the number of GFP^+^ cells after C-tDCS (*p* = 0.044). Similarly, the mean GFP^+^-area presented a significant stroke x stimulation interaction in the perilesional region but also in the healthy tissue (F_1,12_ = 0.13, *p* = 0.72; F_1,12_ = 11.11, *p* = 0.006; F_1,12_ = 10.45, *p* = 0.0072, respectively, for the core, perilesional and healthy ROIs), with corrections analysis indicating a reduction in the Stroke/C-tDCS vs. Stroke/sham groups (*p* = 0.014) in the perilesional area and a trend toward reduction in the healthy region (*p* = 0.094).

### 3.3. C-tDCS after Stroke Modulates Microglial Process Morphology and Phenotype in the Perilesional Ischemic Region

The significantly decreased number of GFP^+^ microglia cells and GFP^+^ area in the perilesional ischemic area of stroke/C-tDCS mice prompted us to investigate whether C-tDCS modulates microglia morphology. For this purpose, we performed a detailed 3D microglial morphology analysis of this region and we observed a more ramified morphology in stroke/C-tDCS compared to stroke-sham mice (Figure 5A). Quantitative morphometric analysis at the level of single cell in the ischemic core, perilesional region and contralateral hemisphere of both groups showed that C-tDCS had no effect on either the cell body sphericity or on the volume and area in any of those regions (Figure 5B–D). Moreover, microglia of stroke/C-tDCS mice found in the perilesional stroke area showed an increase in the total process area, number of process branch points and number of terminal process branches and a trend toward an increase in the process volume compared to the sham group. In contrast, no changes were observed in the healthy region and no analyses were performed in the ischemic core due to the lack of microglial processes (Figure 5E–H).

Taken together, our data showed that application of C-tDCS after stroke enhanced the ramified microglial population exclusively at the perilesional region of the ischemic core area. From this observation, we then explored the possibility microglial phenotype modulation by C-tDCS. We analyzed the number of cells expressing CD68, a characteristic inflammation mediator for phagocytic microglia. We detected a higher number of CD68^+^ cells in the core, with no difference between sham and C-tDCS mice. In the perilesional region, on the contrary, a significantly reduced number of CD68^+^ cells was found, indicating less microglial activation in this region (Figure 6A,B). We next evaluated the expression of the M2 microglial phenotype marker CD206. At our time-point, CD206 immunoreactivity was clearly observed only in the ischemic core, and we observed a significantly higher level of CD206 expression in the C-tDCS group with respect to sham treated mice (Figure 6C,D). In accordance with morphological data, our observation shows a shift in microglia activation toward a more related anti-inflammatory and immunosuppressive response in C-tDCS treated mice.

### 3.4. Loss of NeuN Immunoreactivity after Cerebral Ischemia Is Not Reverted by C-tDCS

The observed improvements in motor behavior and microglia phenotype prompted us to explore whether this early stimulation protocol might also have a neuroprotective effect at 48 h after ischemia. Thus, we performed a staining for NeuN, a specific marker for neurons widely used to identify neuronal loss. Counts of NeuN^+^ cells revealed no differences between sham and C-tDCS mice in any of the analyzed regions. (Figure 7A,B).

## 4. Discussion

It is well established that tDCS can polarity-dependently modulate neural cortical excitability in humans [24,25,26] and, similarly, in animal models [22,27,28]. Moreover, a large body of recent evidence indicates that tDCS is capable of influencing glial cell activity in physiological conditions [19,20,29] and in transient MCAO models [14,30]. Here, in the mouse photothrombotic stroke model, we further investigated the astrocytes and microglia responses on the effects of C-tDCS applied early after preferred forelimb M1 focal ischemic stroke onset (6 h). Due to its small variability in the size and location of the ischemic damage, photothrombosis has been widely used as an experimental model to investigate brain protective mechanisms and cellular responses in vivo [31]. To the final aim, we worked with the Cx3CR1^GFP/+^ mice, in which GFP is expressed in brain microglia under control of the endogenous Cx3cr1 locus, widely used in ischemic stroke studies [8,32].

Initially, we estimated the effects of C-tDCS on the lesion volume using an intermittent protocol that lasts a total of 40 min within an hour, and we found no differences in the total ischemic volume measured at 48 h after stroke onset. This finding is in line with our previous observation, in which the same stimulation protocol starting 4.5 h after the induction of a 90-min MCAO showed no differences in the ischemic volume evaluated at 24 h postischemia, compared with sham stimulated mice [14]. According to different preclinical studies (for a review, see [33]), the stimulation onset with C-tDCS seems to be the determinant in reducing the ischemic lesion only if starting during or shortly after the stroke onset, though the small number of studies and the great variability in the stimulation protocols reported so far make it difficult to reach a clear conclusion on the real effectiveness of early C-tDCS on brain tissue preservation. In the present study, we used similar tDCS intensity (250 μA) as in our previous papers, in which C-tDCS is known to influence neural cortical activity [22,27], but this charge density (33.3 kC/m^2^) does not induce tissue damage and falls within the safety range for preclinical models [34].

Following a motor cortical ischemic attack, persistent disability is a major issue, with one or more body parts contralateral to the infarct impaired or paretic. Loss of function after stroke is due to the directly damaged tissue as well as cell dysfunction in the surrounding areas [5,35]. Forelimb motor deficits contralateral to the ischemic hemisphere were observed 2 days after photothrombosis in the gridwalking and cylinder task, with respect to pre-ischemia behavior (baseline). Of note, significant improvements in motor performance were reported in mice treated with tDCS 6 h after ischemia, with respect to sham animals. The application of C-tDCS revealed an increased spontaneous preferred forelimb use and a lower number of foot-faults. On the contrary, in the open-field test and in the rotarod, no behavioral effects of focal ischemia were observed, suggesting that the focal lesion of a forelimb motor area does not affect locomotion. Thus, this study supports that a specific recovery of motor function of the affected forelimb following stroke can be achieved if applying C-tDCS up to 6 h after permanent ischemia.

A glial event that contributes to the initial response after an ischemic event is astrogliosis, mostly present in the perilesional region surrounding the core, and characterized by reactive astrocytes with high expression of the glial fibrillary acidic protein (GFAP) [36,37]. The lesioned tissue is surrounded by astroglial scar that borders necrotic from healthy tissue, with the undesirable consequence of inhibiting axonal regrowth [38]. Therefore, we investigated the potential role for tDCS in modulating reactive astrogliosis, though we found no effects at 48 h after ischemia and tDCS application. However, the astrogliosis scar is a heterogeneous and variable process and different results might be observed at different time points [39]. Transcranial direct current stimulation has been observed to induce astrocyte activation at least in superficial cortical layers in the physiological state [19], but also to influence the activation of astrocyte in an MCAO model of ischemia in mice, by decreasing GFAP expression and GFAP^+^ cell density and body size in the perilesional region [40].

A second key cellular hallmark after ischemia is the activation of microglia, the resident brain immune cells. Thanks to their ramified morphology, microglia have a surveillance function marked by constant monitoring of the surrounding microenvironment, by dynamically extending and retracting their processes. Ramified cells are characterized by small somas and extensive arborization necessary for active surveillance [41,42]. In response to damage, microglia cells are capable of a variety of morphological changes and strong proliferating activity up to two weeks, with a robust phagocytic capacity within the first two days of stroke [43]. Though the core region is characterized by cell necrosis, surrounding regions are characterized by cellular suffering and different microglia activation that promotes the inflammatory cascade within the brain acutely and affects the neural network remapping in the chronic phase after stroke [44,45]. Importantly, our ischemic mice treated with C-tDCS showed a reduced microglia density, and the total area decreased if compared to non-stimulated mice only in the perilesional region. This result is in line with recent data showing a similar effect in the acute phase after MCAO in rats [40].

After ischemia, a wide morphological variety of microglia is observed. Highly ramified microglia have been found in the hemisphere contralateral to the lesion, while less ramified cells in the perilesional region and amoeboid-shaped cells in the ischemic core [46]. Our 3D reconstruction of microglia and their morphometric analysis revealed that C- tDCS resulted in a significant increase in microglial branching and process area in the perilesional region, without affecting the morphological features of the cell body. Sphericity, cell body volume and area remain constant in tDCS and sham mice, indicating that C-tDCS-induced morphological changes after 6 h are somewhat process-specific. These changes suggest that tDCS would evoke microglial transformation from an active to a more surveillance state, allowing microglial processes to scan more efficiently the microenvironment for restoring brain homeostasis. Though we did not find any difference in the soma morphology after tDCS, the latter has been shown to modulate the body morphology toward a decreased surveillance state without affecting ramified processes when applied to awake mice in physiological conditions [29]. Intriguingly, our findings revealed that tDCS-evoked microglial morphological changes were restricted to the perilesional region, as this effect was absent in both ischemic core and contralateral hemisphere. This was in line with previous studies that consider the perilesional region as potentially salvageable tissue [47]. These observations were also corroborated by the immunoreactivity analysis of the CD68 marker of microglial activation in this area, where we found a decreased CD68 expression evoked by C-tDCS. A similar finding was recently observed in a MCAO rat model in which cortical electrical stimulation at 20 Hz exerted a positive anti-inflammatory effect [48]. At 48 h, the anti-inflammatory marker CD206 was exclusively found in the ischemic core, suggesting that the microglia/macrophages in the ischemic core endorse brain tissue repair and function restoration [49]. Our results showed that CD206 expression was significantly increased in tDCS-treated animals, indicating that it is more committed to protective phenotype cells that are present in the ischemic core, further suggesting a beneficial role of the stimulation. Although the impact of C-tDCS on post-ischemic neuroinflammation was not analyzed in detail, our data show that the application of C-tDCS 6 h after ischemia has a potential anti-inflammatory effect in the injured hemisphere, and microglia might represent a target for this approach in the early phases after acute ischemic stroke.

In parallel with glial activation, neuronal cell death is a rapid process that occurs within the first few hours after stroke [50]. When evaluating neurons’ survival, we did not observe any difference in the number of NeuN^+^ cells. On the contrary, similar studies in which C-tDCS was applied starting from 2 days after MCAO [40] or with deep cortical stimulation [48] demonstrated a clear neuroprotective effect on the rat brain. However, in addition to a different stimulation onset and protocol, our result may also be attributed to a slow clearance rate of degenerating neurons and retention of the NeuN signal in some degenerating neurons [50]; moreover, under pathological conditions, it has been observed that a decrease in NeuN labeling might also reflect a depletion of the protein or loss of its antigenicity [51]. Thus, these results on the neuroprotective role of early tDCS must be considered carefully. Together with this, the study presented here has also some other limitations which may be addressed in future studies, mainly regarding a deeper microglia polarization analysis toward an M1 or M2 profile and an inflammatory cytokine analysis. Moreover, another key point might be to determine both astrocyte and microglia roles at different time-points, specifically during the sub-chronic or the chronic phases after ischemia (i.e., few days or weeks), since it is known that these cells undergo morphological fluctuations in a spatiotemporal manner in different neuropathologies, which may account for different findings in relation to tDCS application after brain ischemia.

## 5. Conclusions

In line with the increasing amount of literature, our data suggest that cathodal tDCS treatment might be a possible therapeutic potential for stroke treatment when applied early after the brain damage over the ischemic region. Our findings show that the stimulation of the affected motor cortical area might be effective up to 6 h after injury, at least from a functional point of view, and microglia might represent a potential therapeutic target.

## Figures and Tables

**Figure 1 biomolecules-12-00588-f001:**
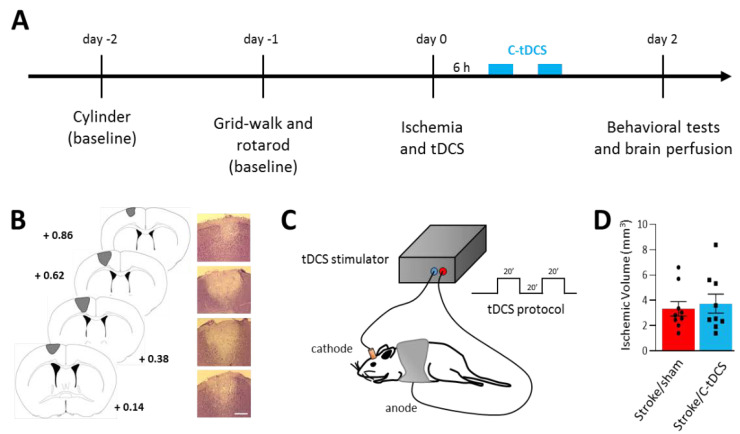
Early application of cathodal tDCS after photothrombotic ischemia in mice. (**A**) Experimental timeline showing the two days of baseline motor tests followed by photothrombotic stroke induction and C-tDCS 6 h afterward. Motor impairments were re-evaluated 48 h after ischemia. (**B**) Representative Nissl-stained brain sections showing the lesion in the M1 area (bar = 300 μm). (**C**) Schematic representation of brain stimulation and C-tDCS protocol consisting of two series of 20 minutes’ stimulation with a 20-min interval. (**D**) Photothrombotic lesion was not different in the sham vs. stimulated groups (black dots and squares, respectively, represent individual values; *n* = 9 per group; *p* > 0.05). Data are given as mean ± SEM (unpaired *t*-test).

**Figure 2 biomolecules-12-00588-f002:**
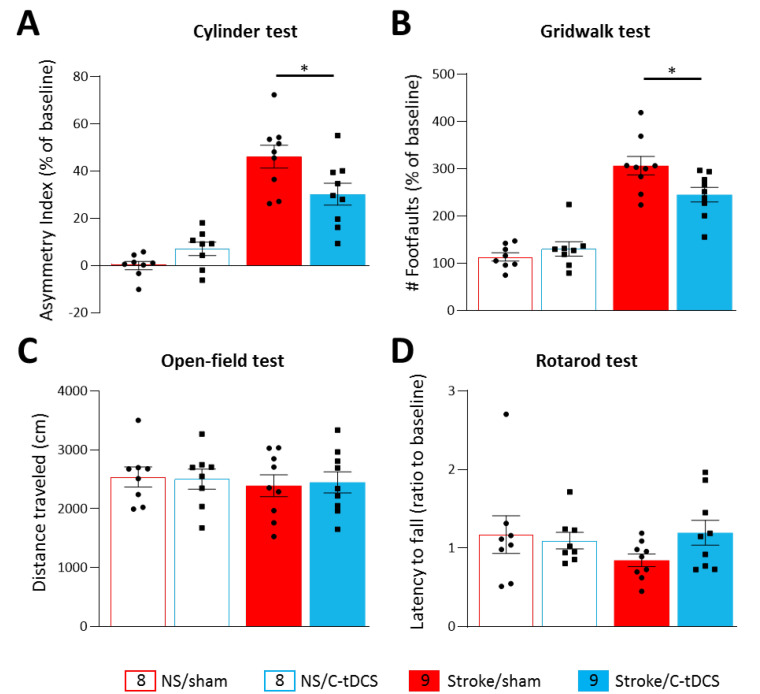
Cathodal tDCS early after ischemia preserves the affected limb’s motor activity. Following photothrombosis, mice showed a significant functional deficit of the affected forelimb in the cylinder (**A**) and the gridwalk test (**B**) with respect to baseline values. The application of C-tDCS in the stroke groups showed a significant reduction in the asymmetry index (*p* = 0.028) and in the number of foot faults (*p* = 0.046) with respect to non-stimulated stroke mice. Motor tests based on locomotor activity and motor coordination showed no differences among the different groups. The OFT analysis of voluntary locomotion (**C**) and the mean latency to fall in the rotarod test (**D**) showed no significant effect of stimulation and photothrombotic ischemia with respect to baseline values. Data are given as mean ± SEM. Black dots (sham) and squares (C-tDCS) represent individual values (*n* = 8 for each no-stroke group; *n* = 9 for each stroke group; two-way ANOVA followed by Tukey’s) * *p* < 0.05.

**Figure 3 biomolecules-12-00588-f003:**
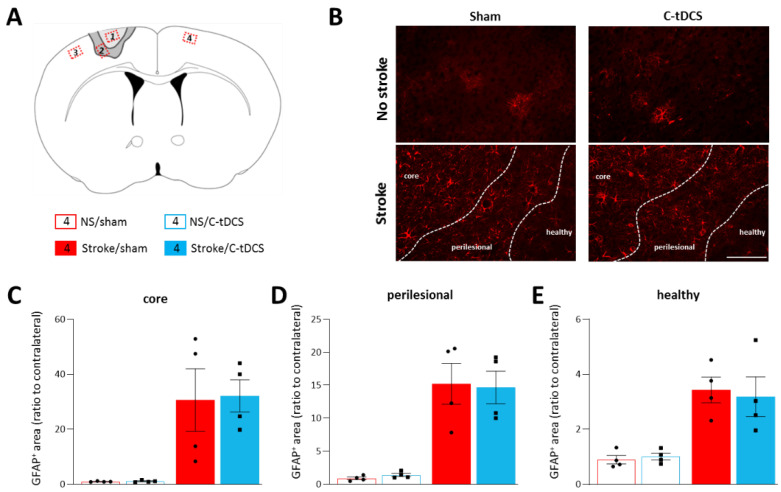
Astrocytes are not modulated by C-tDCS after cortical ischemia. (**A**) Schematic of the areas analyzed with respect to the cortical lesion. Red insets correspond to the analyzed sites (ROIs): 1. Ischemic core; 2. Perilesional region; 3. Healthy tissue in the lesioned hemisphere; 4. Healthy tissue in the contralateral hemisphere. (**B**) Representative images of the four conditions (bar = 100 μm). Quantitative analysis of astrocytes at 48 h after ischemia showed an increased GFAP+ cells in the (**C**) core, (**D**) perilesional area and (**E**) healthy tissue of the photothrombotic hemisphere; the C-tDCS treatment had no effects in any of these regions. Data are given as mean ± SEM. Black dots (sham) and squares (C-tDCS) represent individual values (*n* = 4 per group; two-way ANOVA followed by Tukey’s).

**Figure 4 biomolecules-12-00588-f004:**
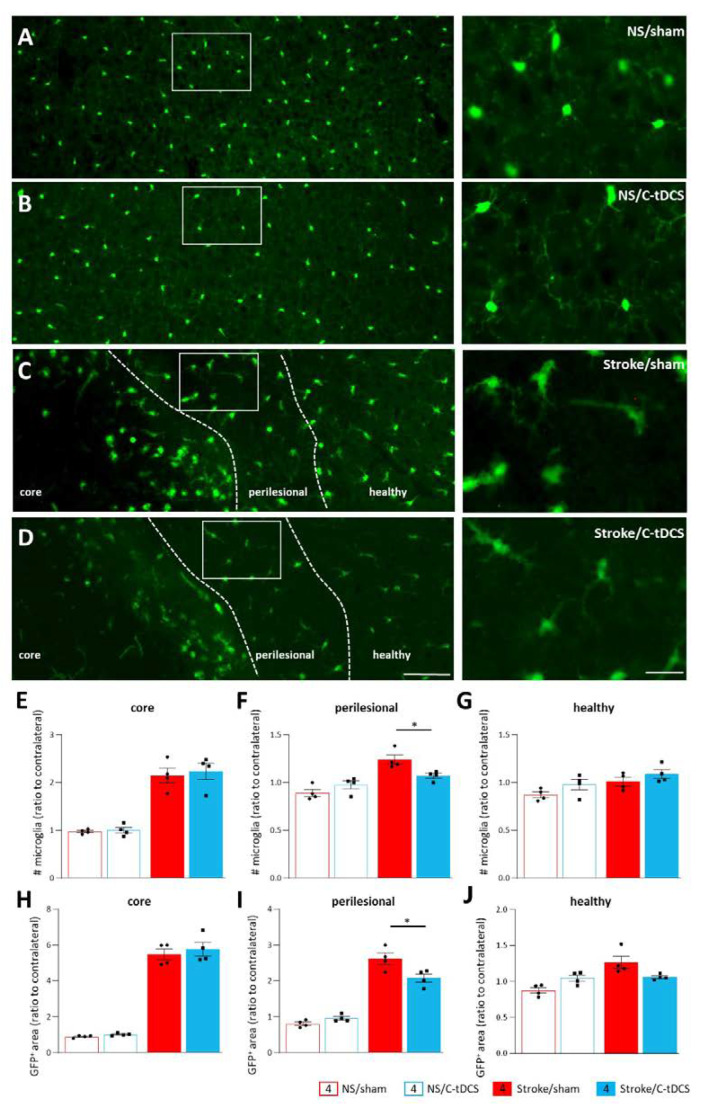
Reduced recruitment of microglia in the perilesional area after early C-tDCS. (**A**–**D**) Representative images showing GFP^+^ microglia in control (no Stroke, NS) and Stroke mice, sham or stimulated with C-tDCS, with magnifications of the perilesional area (right) (bars = 150 and 30 μm, respectively). (**E**–**G**) The number of GFP^+^ microglial cells was higher in the stroke conditions, with no differences in the core and heathy tissue between sham and stimulated mice. In the perilesional region, the number of GFP^+^ microglial cells in the Stroke/C-tDCS was significantly lower with respect to stroke/sham mice (*p* = 0.044). Similarly, the total GFP^+^ area quantification (**H**–**J**) showed a significant decrease in the C-tDCS mice exclusively in the perilesional region (0.014). Data are given as mean ± SEM. Black dots (sham) and squares (C-tDCS) represent individual values (*n* = 4 per group; two-way ANOVA followed by Tukey’s) * *p* < 0.05.

**Figure 5 biomolecules-12-00588-f005:**
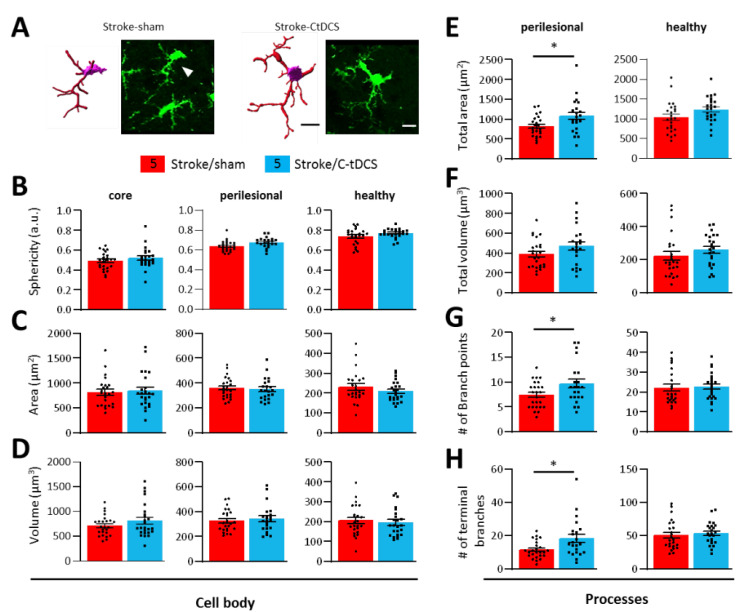
C-tDCS applied after acute stroke induces microglial morphological changes in the perilesional area. (**A**) Representative 3D-reconstructed microglia cells of stroke/sham and stroke/C-tDCS mice in the perilesional stroke area and their relative confocal images (bar = 15 μm). Microglia cells were reconstructed and analyzed from three different areas: ischemic core, perilesional stroke region and contralateral hemisphere (healthy). Quantitative morphometric analysis of microglia cell body (**B**) sphericity, (**C**) area and (**D**) volume) revealed no differences between groups. Analysis of microglial processes showed a significant increase in (**E**) total area (*p* = 0.015) but not (**F**) total volume after C-tDCS, that also resulted in an increased (**G**) number of branch points (*p* = 0.035) and (**H**) terminal branches (*p* = 0.010). All parameters were obtained using Imaris Bitplane software. 4/5 GFP^+^ microglial cells per animal (*n* = 5 per group) were reconstructed and analyzed. Data are presented as a mean ± SEM. Black dots (sham) and squares (C-tDCS) represent individual values (unpaired-*t* test) * *p* < 0.05.

**Figure 6 biomolecules-12-00588-f006:**
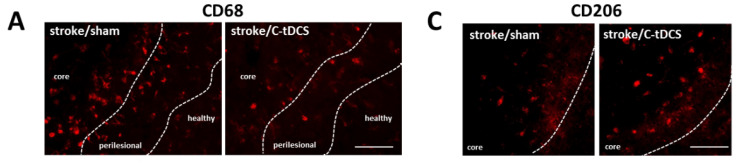
Microglia activation changes following ischemia and C-tDCS early application. (**A**) CD68 immunoreactivity was present in the core, perilesional and healthy regions (bar = 100 μm). (**B**) Statistical quantification of the CD68^+^ area showing a lower expression of the CD86 marker in the perilesional region in the stimulated group (*p* = 0.036). No differences were observed in the core and heathy tissue. (**C**) CD206 expression was evident only in the ischemic core, where a significant increase in the signal was expressed only by C-tDCS mice (**D**; *p* = 0.008). Data are presented as a mean ± SEM. Black dots (sham) and squares (C-tDCS) represent individual values (*n* = 4 per group; unpaired-*t* test) * *p* < 0.05; ** *p* < 0.01.

**Figure 7 biomolecules-12-00588-f007:**
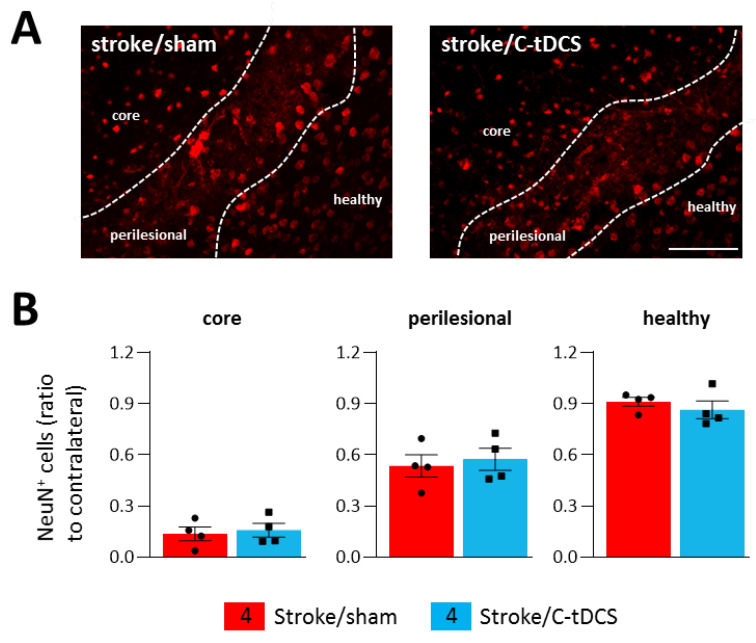
Effects of C-tDCS on NeuN^+^ cell loss after ischemia. (**A**) Representative images of NeuN^+^ cells in stroke mice with and without C-tDCS (bar = 100 μm) revealed no differences (**B**) in the core, perilesional and healthy areas. Data are presented as a mean ± SEM and statistical differences assessed by unpaired-t test.

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
