# Peer review of "Early Application of Ipsilateral Cathodal-tDCS in a Mouse Model of Brain Ischemia Results in Functional Improvement and Perilesional Microglia Modulation"

_biomolecules, 2022, doi:10.3390/biom12040588_

Round 1

Reviewer 1 Report

I think authors performed an interesting and simple approach to understand the cellular origin of the successful C-tDCS treatment. However, in my view the manuscript needs further studies in order to show that the changes observed/described in microglia are the origin of the improvement in functional outcome. Moreover the identification of microglia as the putative origin of the brain protection induced by DCS, has already been shown by other published papers. So authors should clarify the novelty and improve the results obtained regarding microglia role.

- Main findings of the study:

. The authors want to explore the molecular/cellular pathways associated with therapeutic transcranial direct current stimulation (tDCS) in the acute phases after brain ischemia

. Authors explored the effects of cathodal tDCS (C-tDCS) 6 hours after focal forelimb M1 ischemia in Cx3CR1GFP/+ mice.

. Authors found that C-tDCS lead to improved motor functionality in the affected forelimb. At the same time, through histology, they found a decreased microglia cell density in the ischemic area, and changes in morphology, towards a more ramified physiological state

. Finally, authors claim that microglia, might be the key link for the success of C-tDCS treatment in the acute phase stroke

- Limitations

. How was the randomization/blinding between the groups occurred? I did not find any reference to this important issue.

. Any post-operative care performed? Any animal death? Exclusion criteria? See ARRIVE guidelines. Methods should be improved to improve transparency and reproductivity.

. There already different papers in literature that mention microglia role in the tDCS treatment (e.g.: 10.1186/s12868-020-00570-8 ; 10.3389/fnagi.2021.741168 ; etc). What is the novelty of this work?

Major revisions:

. All the points raised in Limitations section that refer to important issues. They have to be addressed, and corrected, when possible.

. Why neuronal population were not analyzed? Even to understand how microglia might be promoting neuroprotection (either in cell death levels or neurite/spine preservation). This should be done, in my view, to enrich the manuscript and clarify potential role of microglia over neurons. That could explain the better functional outcome.

. Inflammatory cytokine analysis would also an important experiment to perform, to understand microglia action, after treatment

. Another point that could be performed or should have been performed is a longitudinal assessment of functional recovery from 2 days till 7 or 15 or 30 days. To understand the extent of recovery promoted by tDCS

. Representative immunofluorescence images in Fig. 3 should be added

. Introduction and discussion should be improved, regarding tDCS brain protection already known

Minor points:

. Graphs should be converted in Dot plots graphs, to facilitate results interpretation and transparency

Author Response

Comments and Suggestions for Authors

I think authors performed an interesting and simple approach to understand the cellular origin of the successful C-tDCS treatment. However, in my view the manuscript needs further studies in order to show that the changes observed/described in microglia are the origin of the improvement in functional outcome. Moreover the identification of microglia as the putative origin of the brain protection induced by DCS, has already been shown by other published papers. So authors should clarify the novelty and improve the results obtained regarding microglia role.

- Main findings of the study:

. The authors want to explore the molecular/cellular pathways associated with therapeutic transcranial direct current stimulation (tDCS) in the acute phases after brain ischemia

. Authors explored the effects of cathodal tDCS (C-tDCS) 6 hours after focal forelimb M1 ischemia in Cx3CR1GFP/+ mice.

. Authors found that C-tDCS lead to improved motor functionality in the affected forelimb. At the same time, through histology, they found a decreased microglia cell density in the ischemic area, and changes in morphology, towards a more ramified physiological state

. Finally, authors claim that microglia, might be the key link for the success of C-tDCS treatment in the acute phase stroke

- Limitations

. How was the randomization/blinding between the groups occurred? I did not find any reference to this important issue.

  1. We thank the Reviewer for this important comment, since in the first version of the manuscript we just stated it in the section 2.6. In the revised version of the manuscript, we clearly stated the blinding analysis for each test. With regard to randomization, we reported that mice were randomly divided in the four groups after baseline tests (Section 2.2).

. Any post-operative care performed? Any animal death? Exclusion criteria? See ARRIVE guidelines. Methods should be improved to improve transparency and reproductivity.

  1. The post-operative care consisted on the application of triple-antibiotic ointment in the region surrounding the electrode and dental cement. Indeed, the electrode was mounted on the intact skull, a non-invasive procedure for the neural tissue which has been already reported in our previous papers (Cambiaghi et al ., 2010, 2011, 2020, Peruzzotti-Jametti el al., 2013), and since our aim was to monitor glial cell modulation, we did not use cortisone or other treatments which would have compromised our data. No animals were excluded from the analysis, with the exception of 2 mice, both died after photothrombotic induction (one never woke up after surgery; one two hours after surgery). We reported these details in the new version of the manuscript.

. There already different papers in literature that mention microglia role in the tDCS treatment (e.g.: 10.1186/s12868-020-00570-8; 10.3389/fnagi.2021.741168 ; etc). What is the novelty of this work?

  1. The reviewer is right and here we have also cited the most relevant papers on the “tDCS and ischemia” topic. However, in these papers, the electrical stimulation has not been applied either in the acute phase (in Zhang et al., after 2 days) after ischemia (our main aim) or non-invasively (in Wang et al., by using invasive electrodes and with biphasic 20 Hz direct current ES), as we have done in the present study. We commented these outcomes in the discussion section. Thanks to Reviewer’s suggestion, we better clarified our aim in the introduction and its novelty, based on the application of cathodal tDCS in the early phases after ischemia, in our case 6 hours, an important time-point in the therapeutic time window of stroke: “Thus, our aim was to study the effects of the acute application (6 hours after ischemia) of cathodal tDCS (C-tDCS) on motor functionality and microglia activation in a mouse model of ischemic stroke.”.

Major revisions:

. All the points raised in Limitations section that refer to important issues. They have to be addressed, and corrected, when possible.

  1. We addressed all the limitations raised by the Reviewer.

. Why neuronal population were not analyzed? Even to understand how microglia might be promoting neuroprotection (either in cell death levels or neurite/spine preservation). This should be done, in my view, to enrich the manuscript and clarify potential role of microglia over neurons. That could explain the better functional outcome.

  1. We thank the Reviewer for this comment. In the new version of the manuscript, we performed a NeuN staining and reported the results in Figure 7. We observed that C-tDCS applied at 6 hours after ischemia resulted in no differences in the number of NeuN+ neurons. We are aware that this is not a complete and final result on the effect of our protocol on neuroprotection, so we stated this as a limitation of the present study.

. Inflammatory cytokine analysis would also an important experiment to perform, to understand microglia action, after treatment

. Another point that could be performed or should have been performed is a longitudinal assessment of functional recovery from 2 days till 7 or 15 or 30 days. To understand the extent of recovery promoted by tDCS

  1. We recognize that these 2 issues might be important in understanding the effects of early tDCS application after ischemia. However, in this study we intended to focus on short effects at the cellular level. Nevertheless, in complete agreement with the Reviewer, we included these points as limitation/future perspectives.

. Representative immunofluorescence images in Fig. 3 should be added

  1. According to the Reviewer’s suggestion we added representative images for GFAP in Figure 3B.

. Introduction and discussion should be improved, regarding tDCS brain protection already known

  1. In line with the Reviewer comment, we implemented this new version of the manuscript, with a more complete description of the literature on the tDCS and ischemia topic.

Minor points:

. Graphs should be converted in Dot plots graphs, to facilitate results interpretation and transparency

  1. Individual values as dot plots has been added to all graphs in the new version of the manuscript.

Reviewer 2 Report

The manuscript shows the beneficial effects of cathodal tDCS on motor functionality and microglial profile when applied in the acute phase in a mouse ischemia model.

Even though the authors present clear and sound results, in my opinion, the claim “our findings suggest a key role of microglia cells in the fate of brain ischemic stroke” is not supported by the study design.

It is not clear to me the specific need for Cx3CR1GFP/+ mice since no live-imaging experiments were performed. For example, the authors could have used wild type animals and performed standard immunofluorescence staining for microglial cells.

When normalized to contralateral, a constant ~10% reduction in all astrocytic and microglial factors presented (Figures 2 and 3) in the “NS/sham” group is shown. Have the authors evaluated this finding?

Moreover, electrode implantation would be expected to cause acute gliosis (i.e., microglial recruitment and activation) at least to some extent. Did the authors use cortisone, or similar, to avoid the condition?

In addition to morphological analysis, a more comprehensive investigation of the post-ischemic microglial profile would be helpful, e.g., M1 vs M2 molecular markers and a potential phenotypic shift.

Minor: There is mislabeling in groups (Figure 1 legend and Figure 2, panel C).

Author Response

The manuscript shows the beneficial effects of cathodal tDCS on motor functionality and microglial profile when applied in the acute phase in a mouse ischemia model.

Even though the authors present clear and sound results, in my opinion, the claim “our findings suggest a key role of microglia cells in the fate of brain ischemic stroke” is not supported by the study design.

  1. We thank the Reviewer for his comments and we totally agree. Accordingly, we remodulated the sentence as follows: “…our findings suggest a positive role for early C-tDCS after ischemia, which is able to modulate microglia phenotype and morphology in parallel to motor recovery”.

It is not clear to me the specific need for Cx3CR1GFP/+ mice since no live-imaging experiments were performed. For example, the authors could have used wild type animals and performed standard immunofluorescence staining for microglial cells.

  1. We decided to use Cx3CR1GFP/+ mice in order to perform a detailed, faster and reproducible microglial morphometric analysis. Differently from the standard immunofluorescence staining for microglial cells, the employment of Cx3CR1GFP/+ mice allows to label completely and faithfully microglial cells reducing the possibility of missing either entire cells or part of their cell bodies and/or processes. Moreover, immunofluorescence staining might give rise to high background signal making it harder to analyze microglial processes.

When normalized to contralateral, a constant ~10% reduction in all astrocytic and microglial factors presented (Figures 2 and 3) in the “NS/sham” group is shown. Have the authors evaluated this finding?

Moreover, electrode implantation would be expected to cause acute gliosis (i.e., microglial recruitment and activation) at least to some extent. Did the authors use cortisone, or similar, to avoid the condition?

  1. Actually, there might be a link between these 2 pointed observations raised by the Reviewer.

Since we mounted electrodes on the intact skull, a non-invasive procedure for the neural tissue, and our aim to monitor glial cell modulation, we did not use cortisone or other treatments which would have compromised our data. After surgery and dental cement application, we treated the skin surrounding the electrode with triple-antibiotic ointment.

Though we implanted epicranial plastic electrodes as in our previous studies (Cambiaghi et al., 2010, 2011, 2020, Peruzzotti-Jametti el al., 2013), the implant was performed immediately after ischemia (or the no-ischemia condition) and not at least a week before, as formerly performed.

Thus, according to the Reviewer’s comment, a statistical analysis between the contralateral and healthy ROIs was performed for the “GFP+ and GFAP+ area” and “GFP+ cells” We observed that there is a tendency to a decrease in some areas, which is not significant (t-test >0.05 for all the combinations).

In addition to morphological analysis, a more comprehensive investigation of the post-ischemic microglial profile would be helpful, e.g., M1 vs M2 molecular markers and a potential phenotypic shift.

  1. In accordance with the Reviewer’s suggestion, we analyzed the expression of CD68 as a marker of activated pro-inflammatory phagocytic microglia and CD206 as a marker of anti-inflammatory M2 phenotype. New results are now showed in Figure 6. Thanks to this suggestion, we detected a shift toward an M2 phenotype, as evidenced by decreased CD68 expression in the perilesional region and an increased CD206 expression in the ischemic core.

Minor: There is mislabeling in groups (Figure 1 legend and Figure 2, panel C).

  1. We thank the reviewer for this comment; we corrected those typos in the new version of the manuscript.

Round 2

Reviewer 1 Report

Authors were able to improve the manuscript, correcting many of the potential pitfalls indicated by the peer-reviewers. The manuscript is now more clear and the results more transparent, to be reproduced by others.

Reviewer 2 Report

I believe the manuscript has been greatly improved after the revision. The authors have addressed all the comments and have made the necessary changes and additions to the text. Therefore, I recommend the manuscript for publication.